# Comparative Assessment of the Level of Patient Safety Culture between Surgical and Nonsurgical Units in Bulgarian Hospitals

**DOI:** 10.3390/healthcare10071240

**Published:** 2022-07-03

**Authors:** Rositsa Dimova, Rumyana Stoyanova, Miglena Tarnovska, Mladen Doykov, Vesela Blagoeva

**Affiliations:** 1Department of Health Management and Health Economics, Medical University of Plovdiv, 4002 Plovdiv, Bulgaria; ros_dimova@yahoo.com; 2Department of Healthcare Management, Section of Medical Ethics and Low, Medical University of Plovdiv, 4002 Plovdiv, Bulgaria; miglena_tarnovska@abv.bg; 3Department of Urology and General Medicine, Medical University of Plovdiv, 4002 Plovdiv, Bulgaria; mladen.doykov@mu-plovdiv.bg; 4First Department of Internal Diseases, Section of Pneumology and Phthisiatrics, Medical University of Plovdiv, 4002 Plovdiv, Bulgaria; vesela.blagoeva@mu-plovdiv.bg

**Keywords:** healthcare professionals, safety patient culture, e-platforms, HSOPSC

## Abstract

Patient safety culture is a key component of the organizational culture and a critical measure of the quality of healthcare. The aim of this study was to gain an insight into the problems concerning patient safety culture, based on the analysis of data, collected after interviewing healthcare specialists working in surgical and nonsurgical units in selected Bulgarian hospitals. This was a cross-sectional online study using a web-platform and the Bulgarian Version of Hospital Survey on Patient Safety Culture. It was conducted among healthcare workers (*n* = 620) in 2021. The B-HSOPSC incudes 42 scales grouped in 12 different domains. We compared the percentage of positive ratings and outcome dimensions between surgical and other hospital departments with the nonparametric Mann–Whitney U test, χ^2^ tests, Fisher’s Exact Test, and OR. The results showed that there are no statistically significant differences between the ratings on Patient Safety Culture given by the surgical and the nonsurgical staff except for the dimension “Hospital management support for patient safety”. Results from the study highlighted that the most important aspect of hospital patient safety is the shortage of medical staff in both surgical and nonsurgical hospital units. Communication, work shift organization, handoffs and transitions between shifts and among different hospital units, as well as communication with line managers were rated as satisfactory in Bulgarian hospitals.

## 1. Introduction

Patient safety culture is a main concern in hospital surgical treatment units. Surgical units, including operating theatres, anesthesia units, obstetrics, and gynecology wards, differ from the nonsurgical departments. In general, they are well known for their high complexity and hazards and their high potential for adverse events and causing harm to patients. Patient safety culture is a key component of the organizational culture and a crucial measure of the quality of health services. In view of this relation, patient safety affects the quality of healthcare, especially in the surgical hospital units. In the process of medical care delivery in the surgical treatment units, it is necessary to understand in depth the context of patient safety culture, as well as what its impact is on the quality of care and ultimately how it influences the risk to patients’ health [1,2,3,4,5].

Statistical reports show that the incidence of errors or adverse events is variable and depends on the nature of medical care, including performing high-risk invasive procedures; on factors of the work environment in the hospitals; on the patients’ age and comorbidities; on the qualification and experience of the medical staff. It is a well-known fact that surgical procedures carry a significantly higher risk for adverse events and errors, compared to nonsurgical units, regardless of the strict adherence to protocols and preventive measures [6]. At present, despite the significant improvements in the surgical safety knowledge, at least half of the adverse events occur during surgical care [6,7,8,9]. On the other hand, many studies have shown that in surgical units, adverse events or errors are reported more often compared to nonsurgical wards [1,2,3,4,5,10].

One study revealed that the incidence of adverse events ranges between 3.8% and 16%, of which 48% are associated with surgical units [11]. A prospective observational study of general-surgery patients found a similar proportion of patients who suffered complications, deemed potentially attributable to errors (18.1%), but these included mostly low-harm events [1].

In industrialized countries, major complications have been documented to occur in 3–22% of inpatient surgical procedures and the death rate is 0.4–0.8%. However, nearly half of the adverse events in these studies were determined to be preventable [1,12,13,14]. With this concern, a variety of measures and actions that aim to improve surgical safety and reduce the number of surgical deaths and complications to patients have been developed. The WHO Guidelines for Safe Surgery are central to this effort [6,15]. Another similar instrument is the Safe Surgery Saves Lives program, which aims to improve surgical safety and to reduce the number of surgical deaths and complications [6,16]. The application of some approaches and methods (such as prospective risk analysis “bow-tie”, Pareto chart, “bundle care”, and Global Trigger Tool) aims to identify the most common reasons for potential adverse events and errors in different areas of hospitalization and to measure patient safety incident frequency. It also helps in identifying adverse events, quantifying the level of harm from each adverse event, and locating the areas needing improvement in medical organizations [17,18,19]. A crucial aspect of the safety management system (SMS) in many high-risk activities is the use of rules to control behavior, which integrates the organization’s efforts in the identification and management of hazards [7]. This will allow those in charge of developing and implementing care based on protocols (standards, guidelines, protocols, and integrated care pathways) in healthcare to expect that the rules they have developed will be followed and will lead to the right solutions.

However, until now, there have been different instruments to assess different aspects of safety culture in hospitals and in the different medical units [20,21]. In order to make an assessment and cross-cultural comparison of the patient safety culture, the Agency for Healthcare Research and Quality (AHRQ) developed a questionnaire in 2004. The Hospital Survey on Patient Safety Culture (HSOPS) is the most frequently used instrument, and it has been translated and utilized in more than 60 countries [22]. Unfortunately, so far, there is little research of the organizational culture in Bulgarian hospitals in general and, in particular, of patient safety in surgical settings, when compared with other hospital units.

The aim of this study is to present an insight into the problems concerning patient safety culture among healthcare specialists in surgical and nonsurgical units in Bulgarian hospitals.

## 2. Materials and Methods

### 2.1. Study Design

This is a cross-sectional study to assess the level of patient safety culture (PSC) between surgical and nonsurgical units in Bulgarian hospitals across the country.

### 2.2. Participants

The survey includes hospital healthcare specialists from all over the country, including internal medicine, surgery, and intensive care units, as well as obstetrics, pediatrics and intensive care wards, laboratory, radiology, psychiatrics, and other departments. Due to the nature of the survey, the snowball sampling method was utilized at each selected hospital instead of random sampling.

### 2.3. Data Collection and Instrument

The survey was organized as a multistep process—initially, 50 out of a total of 346 multidisciplinary private or public hospitals (representing 14.0% of all healthcare facilities in the country) were randomly selected from all 28 administrative areas in the country. A snowball sampling method was used for sample selection. This method is based on referrals from initially sampled respondents of further participants who are believed to have the characteristics of interest [23]. The snowball sampling method allows less control over potential respondents compared to other methods; therefore, until now, we have no information on how many respondents received the hyperlink to study information about its aims. It is not possible to calculate the response rate. In step 1, the web-based questionnaire was distributed to the respondents with the help of the hospital executives or managers. The managers were provided by post and email with information brochures including a link to the web-based platform allowing access to the online questionnaire (www.rsps.bg, access on 28 May 2022) and a cover letter introducing the purpose and the expected outcomes of the project. In step 2, the managers disseminated the link to the web-based questionnaire among the hospital staff. The next stage included a follow-up reminder phone call to the hospital managers.

Patient safety culture was evaluated using the Bulgarian version of Hospital Survey on Patient Safety Culture (B-HSOPSC). The Bulgarian version of the HSOPSC was validated and approved in our previous studies and was proved to demonstrate good psychometric characteristics [24]. The tool includes 42 items to assess the 12 domains of PSC. Our previous research documented its good psychometric characteristics based on factor analysis, reliability, and item analysis [25]. The participants responded to the items on a five-point Likert scale ranging from “strongly disagree” to “strongly agree” or from “never” to “always”. One of the result variables “number of events reported” and “patient safety grade” was scored by the respondents using a scale ranging from “no events” to “1 to 2 events”, “3 to 5 events”, “6 to 10 events”, and “>10 events”. Another variable (overall patient safety grade, scored from 1 to 5—“failing” to “excellent”) was used as an outcome variable as well. The HSOPSC included both positively and negatively worded items; negatively worded items were reversed. The percentage of positive responses for each item was calculated by adding up the positive scores (agree and strongly agree responses). Eventually, the total PSC score was estimated by computing the scores in the 12 domains, after reverse scoring of the negative questions. To obtain the average scores of each dimension, the items were linearly converted to a scale from 0 to 100 points. Thus, higher scores indicate better PSC. Composite-level scores were computed by adding up the positive scores of the items within a composite scale and dividing by the number of items of that composite scale. Sample characteristics were included, for instance, department type, clinical experience, and working area or unit (Table 1).

### 2.4. Ethics Approval and Consent to Participate

This study received ethics approval from the Medical University’s research ethics committee. The questionnaire surveys were anonymous and voluntary for participants.

### 2.5. Statistical Analysis

The percent positive ratings (PPRs) of the 12 dimensions and the positive proportion of outcome variables (patient safety grade and number of events reported) between surgical departments and other departments were compared with the nonparametric Mann–Whitney U test, χ^2^ tests, Fisher’s Exact Test, and OR. The level of significance of 5% probability (*p* < 0.05) was adopted.

Statistical Package for Social Science (SPSS) version 22.0 was used to conduct the statistical analyses.

## 3. Results

### 3.1. Characteristics of the Participants

Of all participants, 160 (27%) worked in surgical treatment units and 433 (77%) worked in nonsurgical units. A total of 620 respondents completed the B-HSOPSC. Of them: 203 (34.2%) were physicians and the remainder represented other healthcare specialists (*n* = 390, 65.8%). Of all healthcare specialists, 160 (25.8%) were from surgical units and 433 (69.8%) were from other units, with the remainder (4.4%) not indicating their workplace. Healthcare specialists from teaching, state-funded, and municipal hospitals predominated (Table 1). The surgical staff came from the following surgical units: surgery 96 (60.0%), critical care 34 (21.25%), and obstetrics and gynecology 30 (18.75%). Nonsurgical staff was distributed as follows: internal medicine 204 (47.1%), pediatrics 36 (8.3%), psychiatry 32 (7.3%), physiotherapy 45 (10.4%), laboratory 52 (12.0%), imaging 27 (6.2%), emergency 23 (5.3%), and other units 14 (3.2%). In addition, more than 80% of the participants indicated that they have direct interaction or contact with patients (*n* = 525, 88.5%). Table 1 shows the detailed characteristic of the respondents.

In general, regarding the participants’ work history in the current hospital, the majority indicated that they had ≥11 years of experience in the studied hospital in a surgical unit (*n* = 134, 37.4%) or (*n* = 239, 40.3%) in a nonsurgical unit. At least 28 (8.3%) and 48 (8.1%), respectively, had worked <1 year in the study setting and in the unit. The majority of participants indicated that they had not reported any adverse event or error over the last year (*n* = 452, 76.2%), followed by those (*n* = 93, 15.7%) who had reported 1–2 events. 

### 3.2. Overall Percentage of the Positive Response Rates of Respondents (PRRs)

Among all 12 safety culture dimensions, the range of PRRs at the hospital level was from 35.16 to 66.65% (Table 2).

The study results documented that among the three dimensions, “Handoffs and transitions” received the highest rate of positive responses, followed by “Overall perceptions of safety” and “Supervisor/manager expectations and actions promoting safety”. More negative responses were given to dimensions “Teamwork across hospital units”, “Non-punitive response to error”, and “Staffing” (see Table 2).

### 3.3. Percent Positive Ratings (PPRs) by the Surgical Department Staff Compared to Nonsurgical Departments

Table 3 presents a comparison among the different dimensions between the staff from the surgical units and those from the nonsurgical units. Between the two studied groups, the percentage of positive responses at the hospital level varied in the surgical departments from 33.91% in D7 to 69.69% in D10, whereas in the other departments, it was from 36.09% in D7 to 66.51% in D1 (Table 3).

As seen in Table 3, there was no statistically significant difference between the ratings of the surgical staff and those of the nonsurgical staff. The dimension “Hospital management support for patient safety” was the only exception as it received significantly lower ratings from surgical units’ staff.

Regardless of the lack of a statistically significant difference between the ratings of the two groups regarding patients’ safety, the dimensions “Staffing” and “Non-punitive response to errors” received the most negative responses from both groups. On the other hand, the surgical staff gave the highest ratings to “Handoffs and transitions” and “Overall perceptions of safety”, compared to nonsurgical staff, who rated higher “Supervisor/manager expectations and actions promoting safety”, “Handoffs and transitions”, and “Organizational learning-continuous improvement” compared to other dimensions.

The medical staff in the surgical treatment units did not consider that “Hospital management provides a work climate that promotes patient safety” (F1) (OR: 1.898; 95% Cl: 1.313–2.742) or that “The actions of hospital management show that patient safety is a top priority” (F8) (OR: 1.715; 95% Cl: 1.188–2.474), i.e., in two out of the three statements in the “Hospital management support for patient safety” dimension, the surgical staff gave more negative ratings compared with the staff from the nonsurgical units.

Despite the lack of a statistically significant difference between the two groups, regarding the ratings in the different dimensions, some variations in the dimension elements were established.

Regarding patients’ safety, at the hospital level, medical staff from the surgical treatment units gave more negative ratings compared to the staff from the nonsurgical departments. In terms of the line managers in the surgical units, the staff generally did not believe that “My supervisor/manager seriously considers staff suggestions for improving patient safety”—B2 (OR: 1.517; 95% Cl: 1.043–2.206). Most of them considered that “My supervisor/manager overlooks patient safety problems that happen over and over”—B4 (OR: 1.512; 95% Cl: 0.998–2.293). They found themselves not informed: “We are informed about errors that happen in this unit” (C3) (OR: 1.419; 95% Cl: 0.984–2.046), and they did not think that “Mistakes have led to positive changes here” (A9) (OR: 1.543; 95% Cl: 1.068–2.229)

The medical staff in the surgical treatment units considered that they “do not have enough staff to handle the workload“ (A2, (OR: 1.451; 95% Cl: 0.983–2.142) and that they do not work as a team: “When a lot of work needs to be done quickly, we work together as a team to get the work done“ (A3), compared with the staff working in the nonsurgical departments (OR: 1.424; 95% Cl: 0.989–2.050). Group variations were also established in the ratings of “We work in ‘crisis mode’, trying to do too much, too quickly“ (A14). To this question, the surgical staff gave more negative ratings compared to the nonsurgical units (OR: 1.508; 95% Cl: 0.936–2.428).

The comparison between the surgical and nonsurgical departments in terms of the “Patient Safety Grade” and the “Number of Events Reported” revealed no statistically significant difference in “Patient safety grade” (E1) and in “Number of events” between the two studied groups (*p* > 0.005).

The data analysis showed no statistically significant difference between the two groups regarding the respondents’ overall ratings and the level of safety at the workplace (*p* > 0.005) (Table 4).

The nonparametrical analysis showed no statistically significant difference regarding the overall assessment of patient safety grade in terms of the respondents’ years of medical experience or their academic background (*p* > 0.005). On the other hand, physicians gave lower grades regarding the overall assessment of patient safety grade, compared to other healthcare specialists (*p* = 0.000); similarly, healthcare staff directly involved in patients care provide lower grades in terms of PSC (*p* = 0.001).

## 4. Discussion

Our research is the first attempt to compare the level of safety culture between the surgical departments and the nonsurgical departments in Bulgarian hospitals based on the B-HSOPSC. We compared the percentage of positive ratings and outcome dimensions (patient safety grade and number of events) between the two groups of respondents from surgical departments and nonsurgical departments and highlighted the advantages and disadvantages of each.

It is known from previous research that most adverse events are attributed to surgeons (46.1%, of them—22.3% negligence). Among all surgical adverse events, 5.6% resulted in death, representing 12.2% of all hospital deaths in Utah and Colorado. Moreover, operative adverse events comprised 44.9% of all adverse events; 16.9% were negligence and 16.6% resulted in permanent disability [7,26,27].

Based on the relatively low PRRs in the 12 dimensions, established in our study, those having PRRs exceeding or equal to 65.0% were regarded as areas of strength, whereas those with PRRs less than or equal to 64% were considered as fields of weakness. Authors of similar research works have chosen a significantly higher threshold for the positive ratings: equal or exceeding 75% [28].

Our PRRs differ from those in other similar studies. Dimensions “Teamwork within units” and “Organizational learning and continuous improvement” were rated higher in the study of Wang et al. (2017) [29]. On the contrary: dimensions “Communication openness” and “Hospital handoffs and transitions” received lower ratings compared to our study [29,30].

Moreover, the highest rated dimensions by the surgical and the nonsurgical staff were different in our study compared to other studies. In our study, the highest rated dimensions in surgery units were “Handoffs and transitions” and “Overall perceptions of safety”, whereas in other similar studies, both the surgical departments and nonsurgical units’ strong dimensions included “Teamwork within units” as well as “Organizational learning and continuous improvement” [29]. There were several underlying problems regarding the improvement of surgical safety such as: understaffing, lack of culture of trust, absence of error-reporting systems and learning from mistakes, as well as zero tolerance to errors. These problems are central to our respondents. It is obvious that preventive coping strategies need implementation in order to improve PSC. Among them: standardized adverse event control policies, organized surveillance, control and reporting systems, adequately trained staff, continuing medical education, open communication, etc., are expected to provide better outcomes.

Our results document that “Staffing” and “Non-punitive response to errors” were weaknesses in the surgical departments. Similar to our study, other works have also found low PRRs to the dimensions “Staffing”, “Non-punitive response to errors”, and “Frequency of events reported” [29,31]. The “Staffing” dimension, receiving the lowest PRRs, is also the most serious problem regarding patient safety culture in both surgical and nonsurgical settings according to other studies [28,32,33].

In the present study, surgical units staff considered that “the hospital management do not provide a work climate favoring patient safety” contrary to the considerations of the staff from the nonsurgical units. This fact is evident from the response rates in the dimension “Hospital management support for patient safety“. Additionally, the comparison of PRRs regarding safety culture between surgical and nonsurgical departments showed no statistically significant differences, except for the dimension “Hospital management support for patient safety” (D8).

However, in other research works, comparing the safety culture between surgical and nonsurgical departments in terms of their responses to different dimensions, statistically significant differences were established. Assessments, provided from the surgical departments’ staff of the dimensions “Non-punitive response to errors”, “Supervisor/manager expectations and actions promoting patient safety”, and “Hospital management support for patient safety”, were lower than those from other nonsurgical departments [33].

The results from the present study show that the assessments regarding the “Patient safety grade” (E1) between the two respondents’ groups were similar. The majority of participants believed that the “patient safety grade” was “very good”. Moreover, no statistically significant difference was found between the responses of participants from both groups regarding the “Number of events reported” (G1). However, different findings compared to ours were documented by other authors in similar studies. These studies showed that medical professionals from nonsurgical units give more positive ratings of the patient safety culture, and they seem more inclined to report a higher number of adverse events [29,33].

One third (33.6%) of the respondents in a Portuguese study agree that the surgical services are “excellent” or “very good”, whereas 67.3% of the Bulgarian hospital staff generally believe it [11].

An amount of 23.8% of the respondents from the Bulgarian hospitals were willing to report at least one adverse event or error for a 12-month period; on the other hand, other studies documented higher percentages—46% [29].

Of all our respondents, those who reported 1 to 2 adverse events annually constituted the highest percentage in both surgical and nonsurgical departments—18% and 15%, respectively. Furthermore, our study found that the staff from nonsurgical units reported adverse events or errors more frequently compared to those in the surgical units. The percentage of respondents from both surgical and nonsurgical departments, who have not reported any adverse events, was approximately 72% and 78%, respectively.

Portuguese authors have documented similar findings, with 71.4% of the surgical units staff not reporting any adverse events [11].

Authors from China have obtained different results from ours. They noted a significantly lower percentage of respondents who have not reported any adverse event for a 1-year period—the percentages are distributed as follows: 49.94% for the surgical and 58.84% for the nonsurgical staff [29].

The high percentage of not reporting errors or adverse events in the present study (76.2%) could be explained to some extent by the lack of voluntary anonymous information and communication technology (ICT) for the registration of accidents. Within this platform, practitioners could share information and are able to learn from mistakes in order to avoid them. It should also be noted that the lack of reporting systems for adverse events is also due to: the lack of trust among patients and healthcare providers; the presence of a punitive approach instead of trust and mutual sharing. In our country, the issue of medical errors and their discussion is still a taboo and is interpreted as establishing personal guilt (i.e., at an individual’s level) instead of analyzing the cause of the errors. For this reason, the online questionnaire of patient safety used in this study (the B-HSOPSC (www.rsps.bg, access on 28 May 2022)) did not include information that could reveal the identity of the respondents, their workplace, and the hospital they work in. In an attempt to clarify this issue, an additional question was asked, regarding the willingness of the medical staff to report adverse events and errors without fear of punishment in the context of the available voluntary anonymous reporting system. The results showed that 84% of the respondents were inclined to report adverse events or other information regarding patient safety if they consider that it could present a potential hazard to patients’ health; 12% of the respondents demonstrated their disagreement with this policy and 3% neither agreed nor disagreed. Unlike the Bulgarian healthcare system, in other countries, “Learning from mistakes” is an essential factor for the development of a high safety culture [34,35].

Although most of the respondents admitted that mistakes are a real and significant risk to patients and even if reporting systems exist, it is known that not every error is well noticeable and reported [11].

Reporting systems are indeed an important tool for learning regardless of the estimates that out of 20 errors that have occurred, only one will be declared and known [11,36,37,38].

### Strengths and Limitations

For the first time in Bulgaria, a study of PSC between surgical and nonsurgical units was conducted, using the self-administered online anonymous B-HSOPSC questionnaire. This approach minimizes the likelihood of interviewer bias and protects the confidentiality of participants. Thus, objectivity is improved. The study was part of a university project and the participants took part on a voluntary basis without any financial incentives.

Our study had some limitations. The small sample size could affect the results, and it does not allow generalizations to the whole country. Due to the limited sample size, the differences between the surgical and the nonsurgical units were relatively small even in the case where they were statistically significant. Another limitation is that in order to secure anonymity and a good response rate, questions regarding demographic characteristics including sex and age were excluded, and questions allowing identification of the specific hospital (such as name, brand, and address) were omitted. The larger number of nurse-participants (433) compared to doctors (160) is due to the fact that more nurses than doctors are usually employed in the Bulgarian hospitals. The situation is similar to that in the Portuguese study [11].

Despite these limitations, our results may be helpful especially to the national healthcare policy makers and show the need for implementing a reporting system for medical errors and the need to stimulate the improvement of patient safety culture.

## 5. Conclusions

This study was carried out in order to understand the role of medical professionals and the surgical environment in the building of a strong patient safety culture in our hospitals. 

From the results obtained, it is evident that the most important aspect of hospital patient safety is the shortage of medical staff in both surgical and nonsurgical hospital units. It seems that communication, work shift organization, handoffs and transitions between shifts and among different hospital units, as well as communication with the line managers are rated as satisfactory in Bulgarian hospitals. Generally, both groups of professionals consider their unit as acceptable with regard to patient safety. Furthermore, the results show that the culture of safety in surgical services is similar to that in the nonsurgical services.

## Figures and Tables

**Table 1 healthcare-10-01240-t001:** Work-related characteristic of the study participants (*n* = 593) based on the HSOPSC results.

Work Related Details	Surgical Departments *n* (%)	Other Departments *n* (%)	Total *n* (%)
*Position* Physicians Other healthcare specialists *Total*	45 (28.1) 115 (71.9) 160 (100.0)	158 (36.5) 275 (63.5) 433 (100.0)	203 (34.2) 390 (65.8) 593 (100.0)
*Years in hospital* <1 1–5 6–10 ≥11 *Total*	13 (8.1) 50 (31.3) 28 (17.5) 69 (43.1) 160 (100.0)	36 (8.3) 131 (30.3) 113 (26.1) 153 (35.3) 433 (100.0)	28 (8.3) 124 (30.5) 98 (23.8) 134 (37.4) 593 (100.0)
*Years in the department* <1 1–5 6–10 ≥11 *Total*	15 (9.4) 49 (30.6) 25 (15.6) 71 (44.4) 160 (100.0)	33 (7.6) 128 (29.6) 104 (24.0) 168 (38.8) 433 (100.0)	48 (8.1) 177 (29.8) 129 (21.8) 239 (40.3) 593 (100.0)
*Hospital ownership* Nonspecified State-funded/municipal Private *Total*	9 (5.6) 89 (55.6) 62 (38.8) 160 (100.0)	33 (7.6) 255 (58.9) 145 (33.5) 433 (100.0)	42 (7.1) 344 (58.0) 207 (34.9) 593 (100.0)
*Teaching hospitals* Nonspecified Yes No *Total*	9 (5.6) 113 (70.6) 38 (23.8) 160 (100.0)	36 (8.3) 307 (70.9) 90 (20.8) 433 (100.0)	45 (7.6) 420 (70.8) 128 (21.6) 593 (100.0)
*Direct contact with patients* Nonspecified Yes, often No *Total*	4 (2.5) 150 (93.8) 6 (3.8) 160 (100.0)	10 (2.3) 375 (86.6) 48 (11.1) 433 (100.0)	14 (2.4) 525 (88.5) 54 (9.1) 593 (100.0)
*Number of events reported* No event reports 1–2 events 3–5 events 6–10 events 11–20 events *Total*	115 (71.9) 29 (18.1) 9 (5.6) 3 (1.9) 4 (2.5) 160 (100.0)	337 (77.8) 64 (14.8) 18 (4.2) 9 (2.1) 5 (1.2) 433 (100.0)	452 (76.2) 93 (15.7) 27 (4.6) 12 (2.0) 9 (1.5) 593 (100.0)

**Table 2 healthcare-10-01240-t002:** The overall percentage of positive responses to each dimension, (*n* = 620).

Dimensions	N of Items in the Dimension	PRRs * (%)
D10_Handoffs and transitions	4	66.65
D12_Overall perceptions of safety	4	65.48
D1_Supervisor/manager expectations and actions promoting safety	4	64.84
D2_Organisational learning-continuous improvement	3	64.62
D3_Teamwork within hospital units	4	59.72
D8_Hospital management support for patient safety	3	58.33
D5_Feedback and communication about error	3	56.40
D11_Frequency of event reporting	3	54.62
D4_Communication openness	3	54.09
D9_Teamwork across hospital units	4	54.03
D6_Non-punitive response to error	3	39.62
D7_Staffing	4	35.16
Total	42	56.13

* PRRs—Positive response rates.

**Table 3 healthcare-10-01240-t003:** Comparison of the PRR regarding “patient safety culture” dimensions of B-HSOPSC among study participants (*n* = 593).

Dimensions (D)	Surgical Units		Other Units		*p*
	N	PRR %	N	PRR%	
D1_Supervisor/manager expectations and actions promoting safety	160	62.19	433	66.51	0.094
D2_Organisational learning-continuous improvement	160	61.67	433	66.28	0.148
D3_Teamwork within hospital units	160	57.97	433	61.26	0.340
D4_Communication openness	160	51.25	433	56.35	0.105
D5_Feedback and communication about error	160	52.50	433	58.74	0.092
D6_Non-punitive response to error	160	41.46	433	39.65	0.555
D7_Staffing	160	33.91	433	36.09	0.240
D8_Hospital management support for patient safety	160	51.46	433	61.59	0.005 *
D9_Teamwork across hospital units	160	52.50	433	55.02	0.451
D10_Handoffs and transitions	160	69.69	433	66.28	0.503
D11_Frequency of event reporting	160	57.71	433	53.81	0.286
D12_Overall perceptions of safety	160	65.62	433	66.11	0.995

* *p* < 0.05.

**Table 4 healthcare-10-01240-t004:** Overall assessment of patient safety grade in Surgical and Nonsurgical Units.

Patient Safety Grade	Surgical Departments *n* (%)	Nonsurgical Departments *n* (%)	Total *n* (%)
Missing	0 (0.0)	2 (0.5)	2 (0.3)
Excellent	36 (22.5)	108 (24.9)	144 (24.3)
Very good	69 (43.1)	186 (43.0)	255 (43.0)
Good	44 (27.5)	91 (21.0)	135 (22.8)
Acceptable	8 (5.0)	40 (9.2)	48 (8.1)
Poor	3 (1.9)	6 (1.4)	9 (1.5)
Total	160 (100.0)	433 (100.0)	593 (100.0)

## Data Availability

Not applicable.

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
