# Peer review of "Comparative Assessment of the Level of Patient Safety Culture between Surgical and Nonsurgical Units in Bulgarian Hospitals"

_healthcare, 2022, doi:10.3390/healthcare10071240_

Round 1

Reviewer 1 Report

The topic is interesting and well treated. The english language can be improved. 

Author Response

Dear Reviewer,
We are grateful to your valuable comments on our article, “Comparative assessment of the level of patient safety culture between surgical and non-surgical units in Bulgarian hospitals”.
The language was reviewed and grammatical errors corrected.

Reviewer 2 Report

Thank you for the opportunity to review the manuscript entitled "Comparative assessment of the level of patient safety culture 2 between surgical and non-surgical units in Bulgarian hospitals" by Dr. Dimova and colleagues.   In this manuscript the authors report cross-sectional online study and the Bulgarian Version of Hospital Survey on Patient Safety Culture.  In particular, they focused on differences between surgical and non-surgical units.  They found that most important aspect of hospital patient safety is the shortage of medical staff.  They also advocate for he development of a reporting system for medical errors and events as well as an improved overall culture of safety.

The manuscript will require moderate English language editing.  The spelling and punctuation is adequate but there are sections where the syntax and grammar require improvement.

The manuscript is a meaningful contribution in that it represents the first systematic analysis of culture of safety within Bulgarian hospitals.  Given the breadth of the survey across many hospitals, the sample size is very low - a limitation that is acknowledged by the authors.  Please address the following comments:

1)  Your study did not demonstrate significant differences between surgical and non-surgical units.  Furthermore, outcomes for surgical patients often involve care delivered by health care providers from non-surgical units/teams.  I question whether this distinction is important.  It seems that the manuscript identifies a critical need for improved culture of safety broadly in the hospitals surveyed.  Please elaborate on why this distinction and analysis was important.  Why not simply look at culture of safety more broadly across all hospital units?

2)  What are the next steps for this line of investigation.  Will you repeat the survey with the goal of a larger sample size or will you use the current pilot data to develop culture of safety initiatives.

3)  The snowball method of sampling seems arbitrary.  Are there references that validate this method?

Thank you for the opportunity to participate in this review.

Author Response

Dear Reviewer,

We are grateful to your constructive and valuable comments on our article, “Comparative assessment of the level of patient safety culture between surgical and non-surgical units in Bulgarian hospitals.

In this letter, in response to the comments, we would like to present the amendments and corrections, made to the article’s text in order to comply with the suggested corrections.

Reviewer: #2:  Your study did not demonstrate significant differences between surgical and non-surgical units.  Furthermore, outcomes for surgical patients often involve care delivered by health care providers from non-surgical units/teams.  I question whether this distinction is important.  It seems that the manuscript identifies a critical need for improved culture of safety broadly in the hospitals surveyed.  Please elaborate on why this distinction and analysis was important.  Why not simply look at culture of safety more broadly across all hospital units?

Our response: Thanks for your question.  In response, I would like to state, that a large number of studies have found that serious adverse events and medical errors threatening patient safety occur more frequently in surgical, emergency, obstetrics, intensive and etc. units if compared to other non-surgical or critical care units. No doubt, surgical medical care is more complex and invasive surgical procedures place patients at a higher risk. In addition, surgical and critically ill patients are particularly vulnerable to iatrogenic injuries. We believe, that starting from the surgical units will give us a clearer picture from where we should start and where we shall direct our efforts at improving safety culture.  

Reviewer: #2:    What are the next steps for this line of investigation.  Will you repeat the survey with the goal of a larger sample size or will you use the current pilot data to develop culture of safety initiatives.

Our response: In the future, we plan to continue to assess and compare PSC across Bg hospitals in a nationwide survey as well as to investigate PSC in separate hospitals in the country. Accumulation of sufficient data in this novel for Bulgaria area will help us create and offer better strategies for improvement of PSC.

Reviewer: #2:    The snowball method of sampling seems arbitrary.  Are there references that validate this method?

Our response: Reference â„–24 focuses on the snowball sampling method. A brief description is provided in the text.  

The language was reviewed and grammatical errors corrected.

Best regards,

Authors

Reviewer 3 Report

I congragulate the authors for conducting the study with significant scientific merit.

Author Response

Dear Reviewer,

We are grateful to your valuable comments on our article, “Comparative assessment of the level of patient safety culture between surgical and non-surgical units in Bulgarian hospitals.

Best regards,

Authors

Reviewer 4 Report

The paper is well written. Few comments:

- PSC acronym is not explicitated. The athors used both, acronym and extended version, in the manuscript. Please correct it.

- Did you find any difference between academic or not? Different specialities? Work experience? or something like that?

- Please provide in the discussion some options or strategies to correct the work behaviour

Author Response

Dear Reviewer,

We are grateful to your constructive and valuable comments on our article, “Comparative assessment of the level of patient safety culture between surgical and non-surgical units in Bulgarian hospitals.

In this letter, in response to the comments, we would like to present the amendments and corrections, made to the article’s text in order to comply with the suggested corrections.

Review # 4: PSC acronym is not explicated. The authours used both, acronym and extended version, in the manuscript. Please correct it.

Our response: We explained the acronym with its extended version in the text. 

Review # 4:  Did you find any difference between academic or not? Different specialities? Work experience? or something like that?

Our response: We added and commented on this results in the result section. Please, find out the text in lines 242-248: “The non-parametrical analysis showed no statistically significant difference regard-ing the overall assessment of patient safety grade in terms of the respondents’ years of medical experience or their academic background (Р>0.005). On the other hand, physi-cians gave lower grades regarding the overall assessment of patient safety grade, com-pared to other healthcare specialists (Р=0.000), similarly, healthcare staff directly involve in patients care provide lower grades in terms of PSC (Р=0.001)“.

Review # 4:  Please provide in the discussion some options or strategies to correct the work behaviour.

Our response: We described some coping strategies in the discussion section such as standardized adverse events control policies, organized surveillance, control and reporting systems, adequately trained staff and continuing medical education, open communication and etc. Please, see lines 279-283 of the text.

Dear editor, in the process of reviewing, we found out that two of the literature sources were repeatedly cites. Correction required new numeration of the references after item 20.

The language was reviewed and grammatical errors corrected.

Best regards,

Authors